# LaCore: Laplacian Cohesive Subgraphs for Graph Representation Learning

## Abstract

Dense, cohesive subgraphs are valuable anchors for pooling and interpretation in graph representation learning (GRL), yet exact cliques are too strict and average-density heuristics are hub-biased and unstable. We introduce LaCore, a fast two-phase *Laplacian-smoothed reverse peeling* method that rebuilds the graph in a fixed importance order and scores each *connected* component with a smooth ratio that penalizes within-component degree variation. A simple one-step growth test yields a natural *first-peak* stopping rule, and a degree-concentration certificate links low Laplacian energy to near-uniform internal support, making the selected subgraphs cohesive and interpretable. LaCore preserves the scalability of greedy peeling, running in $O((|V|+|E|)\log|V| + |E|k)$, and is learned-parameter-free when used as a pooling operator. On synthetic planted-subgraph recovery and graph classification benchmarks, LaCore consistently improves downstream GRL metrics. The result is a practical, stable alternative to density-only heuristics that plugs directly into modern GRL pipelines.

## 1 Introduction

Graph representation learning has advanced rapidly with node-, edge-, and graph-level objectives, yet remains sensitive to how local neighborhoods are defined and pooled (Kipf & Welling, 2017; Hamilton et al., 2017; Perozzi et al., 2014; Grover & Leskovec, 2016). Dense, coherent subgraphs serve as robust building blocks for contrastive pretext tasks, hierarchical pooling, and interpretable summaries. Classical maximal cliques are ill-suited for large, noisy graphs: they require complete connectivity and are NP-hard to enumerate at scale. Conversely, purely average-density objectives may over-select hubs without guaranteeing strong per-node support.

**Core idea.** We study a reverse-peeling heuristic that reinserts vertices in a fixed importance order and scores each connected component $C$ with a Laplacian-smoothed ratio

$$S_L(C) = \frac{|C|}{Q(C) + \varepsilon}, \qquad Q(C) = \sum_{(i,j) \in E_C} (d_i - d_j)^2, \tag{1}$$

where $d$ is the internal-degree vector on $C$. The parameter $\varepsilon > 0$ acts as a regularizer: it prevents division by zero when $Q(C) = 0$ and also balances a size–smoothness trade-off. Larger $\varepsilon$ values favor looser, larger components and smaller $\varepsilon$ values favor tighter, more uniform ones. The score increases smoothly as well-supported vertices are added (little increase in $Q$), and drops sharply once the component absorbs heterogeneous neighborhoods (large increase in $Q$). The search is greedy (no backtracking), components are connected by construction, and a simple one-step growth test (5) provides a natural stopping rule at the first peak.

**Contributions.** (i) A greedy reverse-peeling routine with a Laplacian-smoothed ratio score $S_L(C) = |C|/(Q(C) + \varepsilon)$ evaluated on each connected component during reconstruction. (ii) A local growth test and natural stopping rule: $S_L$ increases iff the per-step $\Delta Q$ is below a simple threshold; the resulting $S_L$ trajectory is typically peak-then-drop, which we exploit for selection. (iii) A degree-concentration certificate linking small $Q(C)$ to near-uniform internal degrees (cohesion). (iv) A scalable edge-centric $\Delta Q$ update and Disjoint Set Union (DSU) bookkeeping; the overall complexity remains $O((|V| + |E|)\log|V| + |E|k)$. (v) GRL integrations (pooling seeds, GNN explainability) with consistent gains on synthetic graph tasks as well as popular benchmarks.

## 2 RELATED WORK

**Graph representation learning.** Graph neural networks (GNNs) are the main framework for graph representation learning (GRL). Early models such as GCNs (Kipf & Welling, 2017) and GraphSAGE (Hamilton et al., 2017) introduced scalable message passing, while later extensions used attention (Veličković et al., 2019), diffusion (Gasteiger et al., 2019), and positional encodings (Dwivedi et al., 2020). The Graph Isomorphism Network (GIN) (Xu et al., 2019) matched the Weisfeiler–Lehman test in expressivity. More recent transformer-based architectures, including Graphormer (Ying et al., 2021) and SAN (Kreuzer et al., 2021), leverage global attention and positional encodings.

**Hierarchical pooling.** Pooling supports graph-level prediction. DiffPool (Ying et al., 2018) learns soft cluster assignments, while Graph U-Nets (Gao & Ji, 2019) use top-$k$ pooling and unpooling. SAGPool (Lee et al., 2019) and ASAP (Ranjan et al., 2020) refine node selection via attention or structural priors. These methods build hierarchical representations but do not ensure subgraph cohesiveness. Contrastive methods (Veličković et al., 2019; You et al., 2020) also rely on reliable subgraphs for positives.

**Dense subgraph discovery.** Peeling-based approaches such as $k$-cores and Charikar's densest-subgraph optimize thresholds or average degree and can be hub-biased or non-smooth (Seidman, 1983; Batagelj & Zaversnik, 2003; Charikar, 2000). In contrast, we reverse peel and use the Laplacian quadratic form on the degree signal to smooth the search and provide an explicit one-step growth test with a first-peak stop.

**Spectral methods and Laplacians.** Spectral graph theory links Laplacians with clustering. Algebraic connectivity (Fiedler, 1973), Cheeger inequalities (Shi & Malik, 2000; Chung, 1997), and smoothness objectives motivate many relaxations. We use the Laplacian quadratic form on the internal-degree signal as a smoothness prior that penalizes within-component heterogeneity.

**Comparison with pooling approaches.** Existing graph pooling methods for GRL typically produce clusters by learning parameters and leveraging node features, but they often lack explicit structural guarantees. DiffPool (Ying et al., 2018) and Graph U-Nets (Gao & Ji, 2019) generate clusters without explicit cohesion criteria. LaPool (Noutahi et al., 2019) learns feature-based, centroid-driven soft assignments (encouraged by Laplacian variation and optional distance regularization) and does not enforce discrete connected components. In contrast, LACORE is *learned-parameter-free* and *structure-only*: a single peel–reconstruct pass on the input graph yields interpretable subgraphs anchored by degree support and spectral smoothness that are connected by construction.

**GNN explanations.** Most post-hoc explainers identify an explanatory subgraph by repeatedly querying the *trained* GNN for importance signals. GNNExplainer (Ying et al., 2019) optimizes a per-graph soft edge/feature mask by maximizing mutual information with the model prediction. PGExplainer (Luo et al., 2020) trains an explainer network on GNN embeddings to amortize edge-importance prediction to new graphs. SubgraphX (Yuan et al., 2021) explores the subgraph space via Monte Carlo Tree Search and scores candidates with (approximate) Shapley values. These methods rely on model gradients or predictions, often require multiple queries, and do not guarantee connectivity of the returned mask. In contrast, LACORE is *model-agnostic*: a single peel–reconstruct pass on the input graph yields one connected, degree-balanced subgraph that captures the dense neighborhood structure typically aggregated by message-passing GNNs, without relying on the trained GNN embeddings.

## 3 PRELIMINARIES & NOTATION

**Graphs and Laplacians.** Let $G = (V, E)$ be a simple undirected graph, $n = |V|$, $m = |E|$. For $C \subseteq V$, $G[C]$ is the induced subgraph, $E_C$ its edge set. The *internal degree* of $v \in C$ is $\deg_C(v) = |\{u \in C : \{u, v\} \in E\}|$, and we write the internal-degree vector $d \in \mathbb{R}^{|C|}$ with entries $d_i = \deg_C(i)$. The (combinatorial) Laplacian of $G[C]$ is $L_C = D_C - A_C$, where $D_C$ is the diagonal degree matrix and $A_C$ the adjacency matrix. We denote by $\lambda_2(C)$ the *algebraic connectivity*, i.e., the

second-smallest eigenvalue of $L_C$ (if $G[C]$ is connected, then $\lambda_2(C) > 0$). We use $y \sim u$ to denote that nodes $y$ and $u$ are adjacent within $G[C]$.

**Averages and minima.**  $\bar{d}_C := \frac{1}{|C|} \sum_{i \in C} d_i$ and $\delta_C := \min_{i \in C} d_i$.

**Laplacian energy and smoothed score.**  The *Laplacian energy* of internal degrees is $Q(C) := d^\top L_C d = \sum_{(i,j) \in E_C} (d_i - d_j)^2$. For $\varepsilon > 0$, define $S_L(C) := \frac{|C|}{Q(C)+\varepsilon}$.

**Asymptotics and DSU.**  We use a Disjoint Set Union (Union–Find) data structure with path compression and union by rank, giving amortized time $O(\alpha_{\mathrm{Ack}}(n))$, where $\alpha_{\mathrm{Ack}}(\cdot)$ is the inverse Ackermann function; in practice $\alpha_{\mathrm{Ack}}(n) \le 4$ for any realistic $n$ (e.g., $n = 2^{65536}$) so per-operation cost is effectively constant.

**Degeneracy ordering and orientation.**  The *degeneracy* $k$ of $G$ is $k = \max_{H \subseteq G} \min_{v \in H} \deg_H(v)$; equivalently, no subgraph of $G$ has minimum internal degree $> k$. A *degeneracy ordering* removes a minimum-degree vertex repeatedly ($v_i$ denotes the $i^{\text{th}}$ vertex removed in this order). For the reverse-peeling stage of our algorithm, we add back vertices in the order of $\texttt{addOrder} = (v_n, \ldots, v_1)$. We denote $\texttt{idx}[v]$ as the position of $v$ in $\texttt{addOrder}$ (in other words, if $\texttt{idx}[v] > \texttt{idx}[u]$, then the vertex $v$ will be added back later than vertex $u$). We then orient each edge $\{u, v\}$ as $v \to u$ if $\texttt{idx}[v] < \texttt{idx}[u]$. This yields a $k$-degeneracy orientation, i.e., when adding back vertex $w$ during the reverse-peeling process, $\deg^{\mathrm{in}}(w)$ will be $\le k$ for all $w$. For $u \in V$, define $\mathrm{pred}(u) = \{v : v \to u\}$ and $\mathrm{succ}(u) = \{v : u \to v\}$. We will also use $\deg^{\mathrm{in}}(\cdot)$ and $\deg^{\mathrm{out}}(\cdot)$ with respect to this orientation.

**Prefix-sums over successors.**  For $v \in V$ and a threshold $t \in \{1, \ldots, n\}$, let $\textsc{SumSucc.until}(v, t) := \sum_{\substack{y \in \mathrm{succ}(v) \\ \texttt{idx}[y] < t}} \deg(y)$. This is the formal object implemented in our pseudocode (Alg. 2). We maintain a cache $\texttt{predSum}[\cdot]$ for $\sum_{y \in \mathrm{pred}(\cdot)} \deg(y)$ so that, when processing $u$ (with index $\texttt{idx}[u]$), the current neighbor-degree sum for $v \in \mathrm{pred}(u)$ is $S_v = \texttt{predSum}[v] + \textsc{SumSucc.until}(v, t)$. Likewise $S_u = \sum_{y \sim u} \deg(y)$.

**Components during reconstruction.**  "Component $C$" refers to a connected component of the subgraph induced by already reinserted vertices; DSU tracks these components.

## 4  METHOD: LAPLACIAN-SMOOTHED GREEDY RECONSTRUCTION

We reinsert vertices in reverse peeling order, maintain connected components via DSU, and score each component with the Laplacian-smoothed ratio $S_L(C)$. We then describe a two-phase heuristic and an efficient implementation that ensures scalability.

### 4.1  LAPLACIAN-SMOOTHED SCORING

*Why use a Laplacian-smoothed ratio?* Greedy objectives based on average degree are brittle and hub-biased; a single weak node can cause abrupt changes. We instead optimize the Laplacian energy of internal degrees via

$$S_L(C) = \frac{|C|}{d^\top L_C d + \varepsilon}. \tag{2}$$

[1]

This score changes gradually as nodes/edges are added, enabling a stable peel-and-reconstruct search with simple incremental updates.

The score $S_L(C)$ offers the following practical advantages:

---

[1]Because the Laplacian energy $Q(C) = d^\top L_C d$ is not scale-invariant and grows with the size and density of $C$, $\varepsilon$ must be chosen on the same order as typical $Q(C)$ values for the graphs of interest. In practice we tune $\varepsilon$ per dataset rather than treating it as a universal constant. We additionally find that, once $\varepsilon$ is on the right scale, accuracy varies by less than $0.5$ percentage points across four orders of magnitude in $\varepsilon$; see Appendix A.5.

- **Smooth objective → stable search.** The Laplacian smoothness ratio changes smoothly as nodes/edges move, so the peel-and-reconstruct search doesn't thrash.

- **Robustness to noise/outliers.** The score prefers degree-uniform subgraphs; it will not over-grow around hubs or collapse from a single weakly connected node.

- **Incremental, scalable updates with structure.** The edge-centric update and DSU let us maintain $d^\top L_C d$ during reconstruction in $O((|V| + |E|) \log |V| + |E|k)$ time, which makes our algorithm efficient for large graphs.

- **GRL-ready scoring.** $S_L(C)$ gives a single, smooth number we can compare across graphs to rank reconstruction candidates. LACORE clusters can be used directly as pooling seeds or contrastive positives, and we grow a component only while the one-step test $\Delta Q < (Q + \varepsilon)/|C|$ predicts an increase in $S_L$. By the degree-concentration bound in Sec. 5, small $Q(C)$ implies near-uniform internal degrees, so the selected components are structurally cohesive.

Intuitively, if all nodes in a subgraph have similar internal degrees, the Laplacian energy $d^\top L_C d$ is small, internal degrees concentrate around their mean, and the component behaves as a cohesive, near-regular module.

## 4.2 A Two-Phase Heuristic

We adopt a two-phase heuristic inspired by degeneracy ordering and densest-subgraph algorithms: (i) a *peeling* phase, where nodes are iteratively removed based on their current degree, and (ii) a *reverse reconstruction* phase that adds nodes back and scores each connected component.

---

**Algorithm 1** LACORE: Reverse-peeling with Laplacian-smoothed scoring (conceptual)

---

1: **Input:** $G = (V, E)$, small constant $\varepsilon > 0$.
2: Initialize a min-priority queue with all nodes, keyed by degree; initialize empty stack $\mathcal{R}$.
3: **while** queue not empty **do**
4:    Extract node $u$ with minimum current degree.
5:    Push $u$ onto $\mathcal{R}$; remove $u$ and incident edges; update neighbor degrees in queue.
6: **end while**
7: Initialize Union–Find on $V$; set best $\leftarrow \varnothing$, $S_L^\star \leftarrow 0$.
8: **for** nodes $u$ popped from $\mathcal{R}$ in reverse order **do**
9:    Reinsert $u$; for each already reinserted neighbor $v$, union $u$ and $v$.
10:    For each affected component $C$, compute $d$, $L_C$, and $S_L(C)$ via equation 2.
11:    **if** $S_L(C) > S_L^\star$ **then**
12:        Update best $\leftarrow C$, $S_L^\star \leftarrow S_L(C)$.
13:    **end if**
14: **end for**
15: **Output:** best and its $S_L$.

---

The peeling phase (lines 3–6) costs $O((|V|+|E|) \log |V|)$ from priority-queue updates. In the reconstruction phase (lines 8–14), the most expensive step is evaluating $Q(C) = d^\top L_C d$ (line 12). Appendix A.1 provides an approach that replaces the naïve recomputation of $Q$ with an edge-centric incremental update in a fixed degeneracy orientation; this makes maintaining $Q$ cost $O(|E|k)$ overall. Crucially, the overall two-phase control flow is unchanged; only the local computation of $Q$ is made faster. With this substitution, the total complexity is $O((|V|+|E|) \log |V| + |E|k)$.

## 5 Local Growth and Cohesion Certificates

Let $C_t$ be a connected component produced during reverse reconstruction after $t$ vertex insertions, with Laplacian energy $Q_t = \sum_{(i,j) \in E_{C_t}} (d_i - d_j)^2$ computed via the edge-centric update (Alg. 2). Define $S_L(C_t) = |C_t|/(Q_t + \varepsilon)$ with $\varepsilon > 0$.

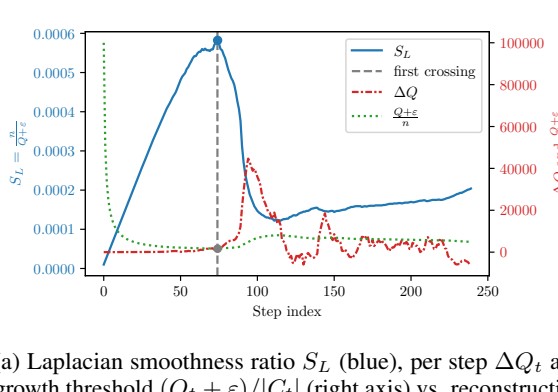 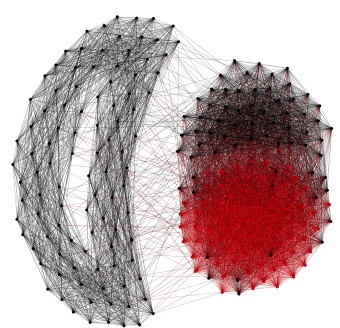

(a) Laplacian smoothness ratio $S_L$ (blue), per step $\Delta Q_t$ and growth threshold $(Q_t + \varepsilon)/|C_t|$ (right axis) vs. reconstruction step for a synthetic graph with a planted dense subgraph. The dashed line marks the first crossing; later crossings may occur but produce smaller bumps and do not affect the global peak selection.

(b) Visualization of the synthetic graph (nodes in gray) and the selected component $C^\star$ (highlighted in red) at the global peak of $S_L$. The dense planted subgraph is correctly identified by LaCore.

Figure 1: Diagnostics for LaCore on a synthetic graph with a planted dense subgraph. Panel (a) illustrates the evolution of the Laplacian smoothness ratio $S_L$ and the one-step growth test, while panel (b) highlights the cohesive component selected by LaCore.

**One-step growth test.** When inserting the $(t+1)^{\text{th}}$ vertex (and incident edges) into $C_t$, let $\Delta Q_t$ denote the change in $Q$. Then

$$S_L(C_{t+1}) > S_L(C_t) \quad \Longleftrightarrow \quad \Delta Q_t \; < \; \frac{Q_t + \varepsilon}{|C_t|}. \tag{3}$$

*Proof.* $S_L(C_{t+1}) > S_L(C_t)$ iff $\frac{|C_t|+1}{Q_t+\Delta Q_t+\varepsilon} > \frac{|C_t|}{Q_t+\varepsilon}$, equivalently $(|C_t|+1)(Q_t+\varepsilon) > |C_t|(Q_t+\Delta Q_t+\varepsilon)$, which simplifies to $Q_t+\varepsilon > |C_t|\Delta Q_t$. $\square$

**Connectivity guarantee.** Each $C_t$ is a connected component of the subgraph induced by reinserted vertices: we only add vertices together with incident edges to already reinserted neighbors and maintain components via DSU, so every time we evaluate or select a best $C_t$, it is connected by construction (Algs. 1–2).

**Cohesion certificate (degree concentration).** Let $d \in \mathbb{R}^{|C_t|}$ be the internal-degree vector on $C_t$, $\bar{d}$ its mean, and $L_{C_t}$ the Laplacian. For any connected $C_t$,

$$\max_{v \in C_t} \left| d_v - \bar{d} \right| \; \leq \; \sqrt{\frac{d^\top L_{C_t} d}{\lambda_2(C_t)}} \; = \; \sqrt{\frac{Q_t}{\lambda_2(C_t)}}, \tag{4}$$

where $\lambda_2(C_t)$ is the algebraic connectivity. Thus small $Q_t$ forces the internal degrees to be nearly uniform, a structural notion of cohesion. A formal proof of Eq. equation 4 is provided in Appendix A.2.

**Peak Diagnostics and Stability.** We visualize the typical peak–then–drop trajectory and the growth threshold. Figure 1 shows, in panel (a), the evolution of the Laplacian smoothness ratio $S_L(C_t)$ over reconstruction steps together with $\Delta Q_t$ and the growth threshold $(Q_t + \varepsilon)/|C_t|$, and, in panel (b), the corresponding selected component for a synthetic graph with a planted dense subgraph.

## 6 INTEGRATION INTO GRL

### 6.1 LaCore POOLING FOR GRAPH CLASSIFICATION

**Pooling operator.** We integrate LaCore as a *learned–parameter–free*, *algorithmic* hierarchical pooling layer: graphs are partitioned via iterative peeling in which LaCore clusters, scored by

$S_L(C)$ (Eq. 2), are sequentially extracted from the remaining graph. This process repeats until either a target node coverage ratio is reached or no clusters of minimum size can be found. Remaining nodes are assigned as singletons. Clusters are contracted to supernodes using mean aggregation, followed by global mean readout.

**Backbone and evaluation protocol.** For our experiments, we employ a standard 2-layer GCN encoder with a LACORE pooling stage inserted between the two GCN layers; we concatenate global mean/max pooled features before and after pooling and feed them to a 2-layer MLP head ($4h \rightarrow 2h \rightarrow C$) with dropout. We train with the Adam optimizer and implement our model in PyTorch. In addition, we follow the protocol popularized by recent pooling work: (i) 10-fold cross-validation, (ii) per-fold validation split of $10\%$ of the training fold, (iii) early stopping on validation loss (patience, up to 500 epochs), (iv) 20 random seeds (we reseed both model initialization and fold generation), and (v) report mean $\pm$ standard deviation over $20 \times 10 = 200$ total test evaluations per dataset.

**Hyperparameter selection.** For each dataset we perform a small grid search on the training–fold validation split and select by validation accuracy. We tune GCN hyperparameters (hidden size, dropout, learning rate, weight decay, batch size) as well as $\varepsilon$ (log grid), coverage target (pooling ratio), and minimum size for LACORE. We apply the same hyperparameter grid search to all other baselines to ensure a fair comparison. For LACORE, we list the final chosen values per dataset in Appendix A.6.

## 6.2 LACORE AS A MODEL-AGNOSTIC GNN EXPLAINER

**Why dense regions explain GNNs.** Graph neural networks compute node and graph representations by aggregating information over local neighborhoods. Their predictions often depend on dense, internally coherent regions where repeated message passing reinforces class-consistent signals. These regions tend to be degree-balanced and structurally stable under low-pass aggregation, and their removal can lead to a significant drop in model prediction accuracy.

**LACORE construction.** LACORE identifies exactly this type of subgraph. It selects the connected component $C^\star$ that maximizes the Laplacian score $S_L(C)$ (defined in Eq. 2). This objective favors dense, hub-averse subgraphs with low degree variance, which preserve signal consistency under aggregation and contribute significantly to the model's prediction. The resulting explanation $C^\star$ is computed without access to gradients, logits, or embeddings, and can be evaluated post hoc by measuring fidelity (the model's prediction change when $C^\star$ is removed).

## 7 EXPERIMENTS

We report clustering/pooling metrics on synthetic graphs, GRL downstream performance on popular benchmark graph datasets, and fidelity–sparsity curves for GNN explainability. The hyperparameter $\varepsilon$ is tuned for each experimental regime, as detailed below.

### 7.1 SYNTHETIC PLANTED CLUSTER RECOVERY

We evaluate LACORE by generating synthetic graphs with $n = 2{,}500$ nodes containing a planted cluster of size $k$, internal edge probability $p_{\text{in}}$, and background probability $p_{\text{out}}$. We sweep $k \in \{100, 150, 200\}$, $p_{\text{in}} \in \{0.6, 0.7, 0.8, 0.9\}$, and $p_{\text{out}} \in \{0.25, 0.35, 0.45\}$, yielding 36 configurations total. Each method is run on 10 random seeds per configuration; seeds are averaged within each configuration before aggregating across settings.

**Baselines and metrics.** We compare against several strong baselines. **Densest Subgraph** uses Charikar's peeling algorithm (Charikar, 2000) to find the subgraph with maximum average degree. **QuasiClique** (Tsourakakis et al., 2013) runs a greedy 1-swap local search to optimize edge density. **Spectral Clustering** uses the principal non-trivial eigenvector of the normalized Laplacian; we sweep both signs and select the best prefix of nodes scored by edge density. **k-core** (Seidman, 1983) iterates through core numbers $k$; for each, it computes the k-core, finds its connected components, and selects

Figure 2: Ablation study for $\varepsilon$ on the synthetic planted subgraph recovery task. Performance is evaluated across a logarithmic scale of $\varepsilon$ values. The peak performance occurs near $\varepsilon = 10^6$ (see text).

the component with the highest edge density. The component with the maximum density found across all $k$ is returned. For fairness, we enforce $|C| \geq 10$ across all baselines; for ranking-based methods (Densest Subgraph, QuasiClique, Spectral) we choose the best candidate satisfying this constraint. Performance is measured by F1-score against the planted set $C^\star$.

**Tuning $\varepsilon$.** The score $S_L(C) = |C|/(d^\top L_C d + \varepsilon)$ balances degree-uniformity (small $d^\top L_C d$) against size ($|C|$). The choice of $\varepsilon$ is critical. Small $\varepsilon$ values heavily penalize non-uniform degrees, leading to small, highly regular subgraphs (high precision, low recall). Conversely, large $\varepsilon$ values diminish the penalty, favoring larger but potentially less coherent subgraphs (high recall, low precision).

Crucially, the magnitude of the Laplacian term $d^\top L_C d$ is not scale-invariant and grows with graph size and density. Consequently, the optimal value of $\varepsilon$ is not a universal constant but depends on the properties of the graphs being analyzed. For any given application domain, $\varepsilon$ should be treated as a key hyperparameter and tuned on a validation set.

To establish a robust value for the synthetic benchmark, we performed the ablation study shown in Figure 2. We generate a separate, fixed validation set for this family of graphs and find that performance peaks near $\varepsilon = 10^6$. For low $\varepsilon$ values, the algorithm becomes too conservative, while for larger $\varepsilon$ values, the regularization becomes too weak. Based on this study, we fix $\varepsilon^\star = 10^6$ for all 36 configurations within this experimental regime. This is the value used to generate the results in Table 1.

**Results.** Table 1 summarizes the results; we report the F1-score, the average performance rank (lower is better), the number of configurations where a method won (achieved the top F1-score), and the median runtime. LACORE consistently outperforms all baselines, winning in every configuration. This advantage is statistically significant: a Wilcoxon signed-rank test on the F1 scores across the 36 settings yields $p < 10^{-6}$ for all pairwise comparisons against baselines. We find that baselines that optimize for average degree perform poorly in this regime because the relatively dense background ($p_{\text{out}} > 0.1$) obscures the planted cluster, which is a fundamental challenge for these heuristics. For a detailed breakdown of performance versus $p_{\text{out}}$, see Appendix A.3

### 7.2 GRAPH CLASSIFICATION BENCHMARKS

**Datasets.** We evaluate on four widely used datasets for measuring graph classification performance: **D&D**, **PROTEINS**, **NCI1**, and **NCI109**, which are taken from the TUDataset collection Morris et al. (2020). Detailed statistics for each dataset are deferred to Appendix A.4. All results use the protocol described in Section 6.1.

**Baselines and families.** We compare to strong representatives from five families: (i) Flat/global pooling (GCN (Kipf & Welling, 2017), Set2Set (Vinyals et al., 2016), SortPool (Zhang et al., 2018), Global-Attention (Li et al., 2016), GMT (Baek et al., 2021)); (ii) Algorithmic coarsening (Graclus

Table 1: Synthetic planted subgraph recovery. Metrics are macro-averaged over 36 graph configurations. LaCORE achieves the best F1-score and average rank over all baselines, with comparable speed to k-core and QuasiClique (Wilcoxon signed-rank, $p < 10^{-6}$). Bold = overall best.

| Method | F1 (macro $\pm$ 95% CI) | Avg Rank $\downarrow$ | Wins | Runtime (ms) [IQR] |
|---|---|---|---|---|
| **LaCORE** | **0.861 $\pm$ 0.070** | **1.00** | **36** | 3466[3657] |
| QuasiClique | 0.217 $\pm$ 0.033 | 2.69 | 0 | 5622[2126] |
| Spectral Clustering | 0.162 $\pm$ 0.017 | 3.33 | 0 | 354[917] |
| Densest Subgraph | 0.113 $\pm$ 0.010 | 3.99 | 0 | **39[21]** |
| k-core | 0.113 $\pm$ 0.010 | 3.99 | 0 | 3811[3114] |

Table 2: **Graph Classification Benchmarks** (accuracy %, mean $\pm$ std over 20 seeds $\times$ 10 folds). Bold = overall best

| Method | DD | PROTEINS | NCI1 | NCI109 |
|---|---|---|---|---|
| *Flat / global pooling* | | | | |
| GCN | 71.67 $\pm$ 1.29 | 66.51 $\pm$ 0.26 | 73.89 $\pm$ 0.62 | 73.78 $\pm$ 0.44 |
| SET2SET | 71.53 $\pm$ 0.77 | 72.07 $\pm$ 0.45 | 66.93 $\pm$ 0.78 | 61.01 $\pm$ 2.73 |
| SORTPOOL | 71.85 $\pm$ 0.96 | 73.92 $\pm$ 0.76 | 68.72 $\pm$ 0.98 | 68.51 $\pm$ 0.59 |
| GLOBAL-ATTENTION | 71.34 $\pm$ 0.82 | 71.81 $\pm$ 0.76 | 69.01 $\pm$ 0.42 | 67.86 $\pm$ 0.42 |
| GMT | **78.09 $\pm$ 0.66** | 74.95 $\pm$ 0.85 | 70.28 $\pm$ 0.55 | 69.53 $\pm$ 0.61 |
| *Algorithmic coarsening* | | | | |
| GRACLUS | 71.95 $\pm$ 4.15 | 72.00 $\pm$ 4.19 | 66.49 $\pm$ 2.39 | 65.33 $\pm$ 3.85 |
| QUASI-CLIQUEPOOL | 66.84 $\pm$ 1.34 | 69.95 $\pm$ 1.04 | 72.26 $\pm$ 0.92 | 67.73 $\pm$ 1.20 |
| *Node-selection pooling* | | | | |
| SAGPOOL | 69.76 $\pm$ 0.84 | 72.33 $\pm$ 0.95 | 64.33 $\pm$ 1.03 | 69.86 $\pm$ 1.45 |
| ASAP | 69.86 $\pm$ 0.93 | 73.41 $\pm$ 0.79 | 64.43 $\pm$ 0.42 | 67.68 $\pm$ 0.57 |
| TOPKPOOL | 70.88 $\pm$ 0.89 | 73.14 $\pm$ 1.12 | 61.70 $\pm$ 2.15 | 66.95 $\pm$ 1.81 |
| *Edge-contraction, soft clustering, parsing-based pooling* | | | | |
| DIFFPOOL | 67.17 $\pm$ 2.52 | 68.49 $\pm$ 1.91 | 62.59 $\pm$ 1.97 | 62.27 $\pm$ 1.85 |
| GPN | 77.82 $\pm$ 0.95 | 74.73 $\pm$ 0.82 | **79.97 $\pm$ 0.39** | 77.21 $\pm$ 0.54 |
| MINCUTPOOL | 76.25 $\pm$ 0.81 | 73.48 $\pm$ 1.03 | 75.34 $\pm$ 0.49 | 73.76 $\pm$ 0.53 |
| SEP | 75.58 $\pm$ 0.89 | 73.96 $\pm$ 0.51 | 77.36 $\pm$ 0.27 | 76.12 $\pm$ 0.62 |
| EDGEPOOL | 72.35 $\pm$ 4.07 | 74.31 $\pm$ 4.14 | 71.54 $\pm$ 2.09 | 67.41 $\pm$ 2.46 |
| **LaCORE (ours)** | 76.85 $\pm$ 0.64 | **75.73 $\pm$ 0.42** | 77.10 $\pm$ 0.56 | **77.48 $\pm$ 0.61** |

(Dhillon et al., 2007), Quasi-CliquePool (Ali et al., 2023)); (iii) Node-selection pooling (SAGPool (Lee et al., 2019), ASAP (Ranjan et al., 2020), TopKPool (Gao & Ji, 2019)); (iv) Edge-contraction and soft clustering (DiffPool (Ying et al., 2018), MinCutPool (Bianchi et al., 2020), SEP (Wu et al., 2022), EdgePool (Diehl, 2019)); (v) Parsing-based pooling (GPN (Song et al., 2024)). For each baseline we use the PyTorch Geometric implementation if available, otherwise the authors' provided code.

**Results.** LaCORE pooling delivers a new high on **PROTEINS** and **NCI109** while remaining competitive on D&D/NCI1 against recent learned/global pooling methods. We attribute this to the *low–variance structural prior* of our pooling: maximizing $S_L$ promotes near–regular, cohesive modules with high minimum support, which are robust aggregation units.

### 7.3 GNN EXPLAINABILITY BENCHMARKS

**Setup.** We evaluate explanations on the **MUTAG** (Debnath et al., 1991; Morris et al., 2020) and **BA-2Motifs** (Luo et al., 2020) graph classification datasets (see Appendix A.4 for dataset details). We use a 3-layer GCN, trained to convergence and then frozen for all explanation experiments.

**LaCORE explanations.** For each graph, we compute the LaCORE cluster $C^\star$ once on the raw graph. We sweep $\varepsilon$ in $S_L(C)$ (Eq. 2) to obtain clusters $C^\star(\varepsilon)$ of varying sizes. When decreasing $\varepsilon$

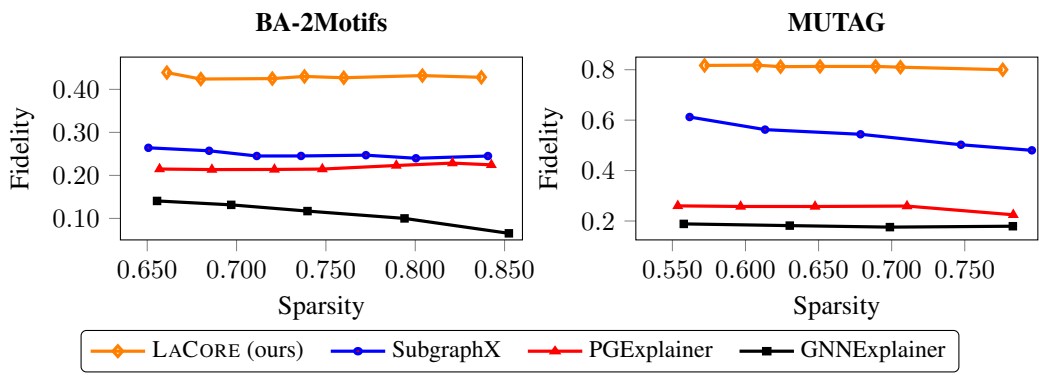

Figure 3: Fidelity vs. sparsity curves on BA-2Motifs and MUTAG datasets. LaCore consistently achieves higher fidelity under similar sparsity levels.

stops reducing the cluster size (e.g., $\varepsilon \leq 10^{-6}$ returns the same cluster), we form smaller explanations of target size $k$ by selecting the top-$k$ nodes in $C^\star(\varepsilon)$ with highest internal degree in $G[C^\star(\varepsilon)]$, denoted $C^\star_k$.

**Baselines.** We use the official DIG (Liu et al., 2021) implementations of GNNExplainer, PGExplainer, and SubgraphX with their default hyperparameters. [2] For each target size $k$, we use the method's built-in control (e.g., `sparsity`, top-$k$ edges, or `max_nodes`) to produce a node set of roughly $k$ nodes; we plot the achieved sparsity for each point.

**Evaluation & Metrics.** Let $p(G) = \text{softmax}(f(G))$ be the class probabilities of the frozen GCN on the original graph, and let $\hat{y} = \arg\max_c p_c(G)$ be its predicted class. For any method's explanation $C_k$, we form the graph $G \setminus C_k$ by deleting those nodes (and incident edges) and compute $p(G \setminus C_k)$. The reported fidelity is Fidelity$(k) = p_{\hat{y}}(G) - p_{\hat{y}}(G \setminus C_k)$, with larger values indicating stronger dependence of the prediction on the removed subgraph. We plot fidelity on the y-axis against the sparsity $1 - |C_k|/|V|$ on the x-axis, for each dataset and method.

**Results.** Across both datasets and all tested sparsities, LaCore attains higher fidelity than GNNExplainer, PGExplainer, and SubgraphX. The gains are most pronounced at higher sparsities (smaller explanations), indicating that the LaCore cluster preserves the model-relevant structure more compactly; see Figure 3.

## 8 LIMITATIONS

The score focuses on internal degree smoothness; other notions (e.g., edge weights, higher-order motifs) could be integrated. Computing $\lambda_2(C)$ exactly is expensive for large $C$; in practice one may rely on proxies or omit it outside the certificate. Additionally, the optimal choice of $\varepsilon$ is sensitive to graph scale and density and may need tuning across domains; nevertheless, within a fixed regime we observe that performance is stable over a broad log-range of $\varepsilon$ (Appendix A.5).

## 9 CONCLUSION

We introduced LaCore, a learned-parameter-free method for discovering cohesive subgraphs through Laplacian-smoothed reverse peeling, optimizing $S_L(C) = |C|/(Q(C)+\varepsilon)$ to balance size and degree uniformity with theoretical guarantees on connectivity and cohesion. Our results reveal a broader principle: structural smoothness provides a robust inductive bias for graph learning that can match or exceed learned approaches while remaining interpretable and scalable. Consistent gains across diverse tasks (planted subgraph recovery, graph classification, and GNN explanation) suggest that

---

[2]`https://github.com/divelab/DIG.`

degree-balanced, cohesive structures are fundamental building blocks that GNNs implicitly seek during training. Looking forward, LACORE's framework naturally extends to weighted graphs and higher-order structures, could anchor graph coarsening for large-scale GNNs or provide interpretable summaries for scientific discovery, and points toward hybrid methods that combine algorithmic guarantees with learned representations.

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

## A  APPENDIX

### A.1  EFFICIENT IMPLEMENTATION AND COMPLEXITY

To make the reconstruction phase in our algorithm scalable, we introduce an edge-centric update scheme that avoids recomputing the Laplacian energy from scratch. This scheme processes nodes in reversed degeneracy order and caches intermediate sums.

**Edge-centric Laplacian update.** When adding an edge $(v, u)$ during reconstruction (where $v \in \text{pred}(u)$), let $d_u$ and $d_v$ be the degrees of $u$ and $v$ *before* the insertion. Let $S_u = \sum_{y \sim u} d_y$ and $S_v = \sum_{y \sim v} d_y$ be the sums of their current neighbors' degrees. The Laplacian energy $Q = \sum_{(i,j) \in E} (d_i - d_j)^2$ changes by

$$\Delta Q = (d_u - d_v)^2 + \underbrace{(2d_u^2 - 2S_u + d_u)}_{\text{increment from } u\text{'s edges}} + \underbrace{(2d_v^2 - 2S_v + d_v)}_{\text{increment from } v\text{'s edges}}. \tag{5}$$

The first term is the new edge's direct contribution. The bracketed terms account for the change in energy over edges already incident to $u$ and $v$. To compute $S_x$ efficiently, we cache sums over predecessors and scan successors, leveraging the low out-degree of the degeneracy orientation. Algorithm 2 details this process.

---

**Algorithm 2** LACORE reconstruction with edge-centric $O(|E|k)$ update

---

1: Compute `addOrder` from peeling. Set `idx`, build $\text{pred}(\cdot)$, $\text{succ}(\cdot)$; sort each $\text{succ}(v)$ by `idx`.
2: Initialize $\deg[\cdot] \leftarrow 0$, $\text{predSum}[\cdot] \leftarrow 0$, DSU for components with per-component $Q \leftarrow 0$.
3: **for** $u$ in `addOrder` **do**
4:    $S_u \leftarrow 0$ {sum of neighbor degrees already attached to $u$}
5:    **for** $v \in \text{pred}(u)$ **do**
6:       $a \leftarrow \deg[u]$, $b \leftarrow \deg[v]$.
7:       $S_v \leftarrow \text{predSum}[v] + \text{SumSucc.until}(v, \text{idx}[u])$.
8:       $\Delta Q \leftarrow (a - b)^2 + (2a^2 - 2S_u + a) + (2b^2 - 2S_v + b)$.
9:       Add $\Delta Q$ to $Q$ of `DSU.find`$(u) \cup$ `DSU.find`$(v)$; update best $S_L$ if needed.
10:     $\deg[\mathbf{u}] \leftarrow \deg[\mathbf{u}]+1$, $\deg[\mathbf{v}] \leftarrow \deg[\mathbf{v}]+1$.
11:     **for** $y \in \text{succ}(u)$ **do** $\text{predSum}[y] \mathrel{+}= 1$. **for** $y \in \text{succ}(v)$ **do** $\text{predSum}[y] \mathrel{+}= 1$.
12:     $S_u \mathrel{+}= \deg[v]$ {after increment}
13:   **end for**
14: **end for**

---

**Practical stopping rule.** During reconstruction of a component $C$, stop appending as soon as $\Delta Q \geq (Q + \varepsilon)/|C|$. We still keep scanning the global stream to update the best component across time, but the per-component early stop can yield speedups on large graphs.

**Complexity.** The peeling phase with a binary heap costs $O((|V| + |E|) \log |V|)$. In reconstruction, every edge is processed once. The total work is driven by two main operations performed for each edge $(v, u)$: updating the `predSum` caches for successors of $u$ and $v$, and computing the neighbor-degree sum $S_v$. Both require iterating through successor lists, which are bounded in size by the graph degeneracy $k$. A worst-case analysis shows that the total work for each of these operations, when summed over all edges, is bounded by $O(|E|k)$. DSU unions contribute a near-linear factor of $O(E \, \alpha(|V|))$. The overall time complexity is therefore dominated by the peeling phase and these reconstruction costs, yielding $O((|V| + |E|) \log |V| + |E|k)$.

## A.2 PROOF OF THE COHESION CERTIFICATE (EQ. 4)

Recall that for a connected component $C_t$ with internal-degree vector $d \in \mathbb{R}^{|C_t|}$, mean $\bar{d}$, Laplacian $L_{C_t}$, and algebraic connectivity $\lambda_2(C_t)$, Eq. equation 4 states that

$$\max_{v \in C_t} |d_v - \bar{d}| \leq \sqrt{\frac{d^\top L_{C_t} d}{\lambda_2(C_t)}} = \sqrt{\frac{Q_t}{\lambda_2(C_t)}}.$$

*Proof of equation 4.* Let $\mathbf{1} \in \mathbb{R}^{|C_t|}$ denote the all-ones vector, and define

$$z := d - \bar{d}\mathbf{1}.$$

By definition of the mean, $z$ is centered:

$$\sum_{v \in C_t} z_v = \sum_{v \in C_t} d_v - |C_t|\bar{d} = 0,$$

so $z$ is orthogonal to $\mathbf{1}$.

Because $G[C_t]$ is connected, the Laplacian $L_{C_t}$ is symmetric positive semidefinite with eigenvalues

$$0 = \lambda_1(C_t) < \lambda_2(C_t) \leq \cdots \leq \lambda_{|C_t|}(C_t),$$

and its nullspace is spanned by $\mathbf{1}$. For any vector $x$ orthogonal to $\mathbf{1}$, the Rayleigh-quotient bound (Poincaré inequality) gives

$$x^\top L_{C_t} x \ \geq \ \lambda_2(C_t) \, \|x\|_2^2. \tag{6}$$

We now apply this with $x = z$. First note that

$$L_{C_t} \mathbf{1} = 0 \quad \implies \quad d^\top L_{C_t} d = (d - \bar{d}\,\mathbf{1})^\top L_{C_t} (d - \bar{d}\,\mathbf{1}) = z^\top L_{C_t} z.$$

By equation 6,

$$z^\top L_{C_t} z \ \geq \ \lambda_2(C_t) \, \|z\|_2^2 \quad \implies \quad \|z\|_2^2 \ \leq \ \frac{z^\top L_{C_t} z}{\lambda_2(C_t)} = \frac{d^\top L_{C_t} d}{\lambda_2(C_t)} = \frac{Q_t}{\lambda_2(C_t)}.$$

Finally, we relate the maximum deviation of internal degrees to the $\ell_2$-norm of $z$:

$$\max_{v \in C_t} |d_v - \bar{d}| = \|z\|_\infty \ \leq \ \|z\|_2 \ \leq \ \sqrt{\frac{Q_t}{\lambda_2(C_t)}}.$$

This is exactly Eq. equation 4, completing the proof. $\qquad\square$

### A.3 F1 vs $p_{\text{OUT}}$ ACROSS METHODS

To visualize heterogeneity across regimes, we plot F1 vs $p_{\text{out}}$ averaged over $k$ and $p_{\text{in}}$ with shaded 95% CIs; one line per method.

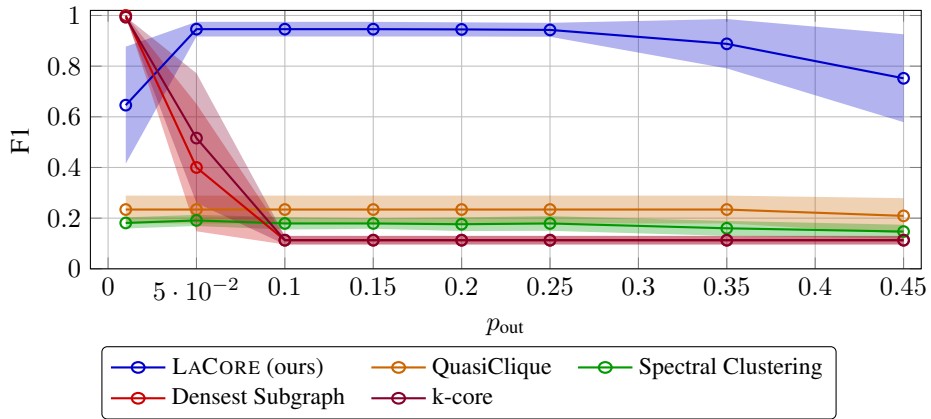

Figure 4: F1 vs $p_{\text{out}}$ (averaged over $k$ and $p_{\text{in}}$). Shaded bands show 95% CIs. The baselines are fully described in Section 7.1. Spectral Clustering uses the principal non-trivial eigenvector; Densest Subgraph is Charikar's peeling; k-core reports the densest component; QuasiClique optimizes edge density.

### A.4 DATASET STATISTICS

Table 3 summarizes the key statistics for the six graph classification datasets used in our experiments. All datasets represent binary classification tasks. PROTEINS contains protein structures classified as enzymes or non-enzymes. D&D consists of protein structures classified by their fold type. NCI1 and NCI109 contain chemical compounds screened for activity against two different types of cancer. MUTAG comprises nitroaromatic compounds labeled by their mutagenic effect on *Salmonella typhimurium*. BA-2Motifs is a synthetic benchmark in which each Barabási–Albert base graph is augmented with either a "house" motif or a 5-cycle; the graph label indicates which motif is attached.

Table 3: Statistics for graph classification datasets. All datasets are binary classification tasks.

| Dataset | Graphs | Avg Nodes | Avg Edges | Classes |
|---|---|---|---|---|
| PROTEINS | 1,113 | 39.06 | 72.82 | 2 |
| D&D | 1,178 | 284.32 | 715.66 | 2 |
| NCI1 | 4,110 | 29.87 | 32.30 | 2 |
| NCI109 | 4,127 | 29.68 | 32.13 | 2 |
| MUTAG | 187 | 18.03 | 39.80 | 2 |
| BA-2Motifs | 1,000 | 25.00 | 51.00 | 2 |

## A.5 $\varepsilon$ SENSITIVITY ON TUDATASETS

To quantify how sensitive LACORE is to the regularization parameter $\varepsilon$ in the graph classification regime, we perform an additional sweep on two TU benchmark datasets, PROTEINS and D&D. We deliberately focus on this pair due to their substantial difference in graph size and density: PROTEINS consists of relatively small graphs with tens of nodes, whereas D&D contains much larger graphs with hundreds of nodes and substantially more edges on average (Table 3). For each dataset we fix all other hyperparameters to the configuration in Table 6 and vary $\varepsilon$ over $\{10^{-3}, 10^{-2}, 10^{-1}, 10^0, 10^1, 10^2\}$, rerunning the evaluation protocol of Section 7.2.

Table 4: $\varepsilon$ sensitivity on PROTEINS and D&D (accuracy %, mean $\pm$ std).

| $\varepsilon$ | PROTEINS | D&D |
|---|---|---|
| $10^{-3}$ | $75.69 \pm 0.27$ | $76.13 \pm 0.57$ |
| $10^{-2}$ | $75.72 \pm 0.40$ | $76.61 \pm 0.73$ |
| $10^{-1}$ | $75.85 \pm 0.31$ | $76.91 \pm 0.86$ |
| $10^0$ | $75.83 \pm 0.29$ | $76.49 \pm 0.68$ |
| $10^1$ | $75.42 \pm 0.37$ | $76.13 \pm 0.69$ |
| $10^2$ | $74.83 \pm 0.36$ | $75.66 \pm 0.93$ |

The resulting accuracy curves are nearly flat across four orders of magnitude in $\varepsilon$. Aggregating over all values in the sweep, the standard deviation of accuracy with respect to $\varepsilon$ is 0.39 percentage points on PROTEINS and 0.44 on D&D. In practice, once $\varepsilon$ is chosen to be on the same scale as typical $Q(C)$ values for a given dataset, performance is largely insensitive to the precise value and $\varepsilon$ behaves as a standard dataset-level hyperparameter that can be tuned coarsely on a validation split.

## A.6 HYPERPARAMETER SELECTION

For each dataset, we performed a grid search over the hyperparameters listed in Table 5. The search was conducted on the training-fold validation split (10% of training data) using 5 random seeds per configuration. We selected the configuration with the highest validation accuracy, which was then used for all baselines to ensure fair comparison. The final chosen hyperparameters for each dataset are shown in Table 6.

Table 5: Hyperparameter search space for graph classification experiments.

| Hyperparameter | Search Space |
|---|---|
| $\varepsilon$ (LACORE) | $\{10^2, 10^1, 10^{-1}, 10^{-2}, 10^{-3}, 10^{-4}\}$ |
| target_ratio (LACORE) | $\{0.25, 0.35, 0.5\}$ |
| min_size (LACORE) | $\{2, 3, 4\}$ |
| batch_size | $\{64, 128, 256\}$ |
| hidden_size | $\{128, 256\}$ |
| learning_rate | $\{10^{-4}, 5 \times 10^{-4}\}$ |
| weight_decay | $\{10^{-3}, 10^{-4}, 10^{-5}\}$ |
| dropout | $\{0.10, 0.15, 0.20, 0.25\}$ |

Table 6: Selected hyperparameters for graph classification experiments.

| Dataset | $\varepsilon$ | target_ratio | min_size | batch | hidden | lr | wd | dropout |
|---------|-----|-------------|----------|-------|--------|------|------|---------|
| PROTEINS | 0.1 | 0.25 | 4 | 128 | 128 | 5e-4 | 1e-3 | 0.10 |
| D&D | 0.1 | 0.25 | 4 | 128 | 128 | 5e-4 | 1e-3 | 0.25 |
| NCI1 | 0.1 | 0.25 | 3 | 64 | 256 | 5e-4 | 1e-3 | 0.20 |
| NCI109 | 0.1 | 0.25 | 3 | 64 | 128 | 5e-4 | 1e-3 | 0.15 |

## A.7 HARDWARE AND IMPLEMENTATION DETAILS

All synthetic planted-cluster experiments were run on a single machine equipped with an AMD Ryzen 7 9700X 8-Core Processor. The software stack consisted of Python 3.12, PyTorch 2.8.0, and PyTorch Geometric 2.6.1 Fey & Lenssen (2019); Fey et al. (2025).

