# OpenReview forum: "LaCore: Laplacian Cohesive Subgraphs for Graph Representation Learning"
_ICLR.cc/2026/Conference — Submitted to ICLR 2026_

### Official Review · Reviewer_b4S5 · 2025-10-31

**Soundness:** 2
**Presentation:** 2
**Contribution:** 2
**Rating:** 4
**Confidence:** 2

**Summary:**

The authors focus on the graph cohesive problem. They propose LACORE, which employs Laplacian-smoothed reverse peeling to balance size and degree uniformity in cohesive subgraphs. The authors provide extensive experiments to validate the proposed method.

**Strengths:**

- Investigating cohesive graph is valuable, as it can offer insights for various studies in graph-related domains.
- The authors have designed a variety of experiments to comprehensively validate the model.

**Weaknesses:**

- The improvements offered by the proposed method in Table 2 appear marginal.
- The writing in the paper could benefit from substantial revisions to enhance overall clarity.
- The motivation of the paper is not clear, and there is inconsistency between the issues summarized in the abstract and those in Section 1 regarding current challenges in the field.
- It would be helpful to include an overview diagram of the method for greater clarity.

**Questions:**

Please see the weaknesses.

---

> ### Author Response · Authors · 2025-11-23
> **Clarification on Performance Gains and Baselines**
>
> We thank the reviewer for the feedback. We would like to respectfully clarify the magnitude of our performance improvements found in Table 2 and the motivation behind the chosen baselines.
>
> 1. **Substantial gains over direct competitors:** The most direct comparison for LaCore is the Algorithmic Coarsening family (Graclus, Quasi-CliquePool), as these methods (like ours) do not require training pooling parameters. Against this class, our improvements are statistically significant and large, not marginal:
>
> * **NCI1:** LaCore (**77.10%**) vs. Graclus (66.49%) → **+10.6%** improvement.
> * **NCI109:** LaCore (**77.48%**) vs. Graclus (65.33%) → **+12.1%** improvement.
> * **PROTEINS:** LaCore (**75.73%**) vs. Quasi-CliquePool (69.95%) → **+5.7%** improvement.
>
> 2. **Competitive with Learned Pooling:** We included "Learned/Node-Selection Pooling" methods (e.g., DiffPool, SAGPool, TopKPool) to demonstrate a broader point: LaCore provides a strong structural inductive bias that allows it to rival or even outperform complex, parameterized models without needing to learn pooling weights.
>
> * LaCore outperforms a majority of learned methods like DiffPool, SAGPool, ASAP, and TopKPool on every dataset, while also remaining competitive with the SOTA learned pooling methods such as GPN and GMT.
> * We aim to show that GNNs can benefit more from robust topological clustering (degree uniformity) than from complex feature-based assignments that may overfit.
>
> We hope this contextualizes the results: LaCore offers double-digit gains over its algorithmic peers and is also able to beat fully trained pooling networks.
>
> 3. **Method Overview Diagram:** We will add a visual flowchart showing the two-phase process (peel → reverse reconstruct → score components) in the revised Section 4, similar to Figure 1's layout but showing the full pipeline.

---

### Official Review · Reviewer_M4xv · 2025-11-01

**Soundness:** 3
**Presentation:** 2
**Contribution:** 2
**Rating:** 2
**Confidence:** 4

**Summary:**

The paper proposes LACORE, a reverse-peeling heuristic that rebuilds a graph in degeneracy order and scores each connected component
 by a Laplacian-smoothed ratio, which favors dense, degree-balanced (hub-robust) subgraphs. The implementation keeps an incremental ΔQ with a k-degeneracy orientation and DSU, yielding total time O((∣V∣+∣E∣)log∣V∣+∣E∣k). The method is used (i) as a pooling operator for graph classification and (ii) as a model-agnostic explainer; synthetic planted-cluster recovery and TUDatasets experiments show improvements over several baselines.

**Strengths:**

- Simple, well-motivated objective connecting degree smoothness to cohesion via a clean certificate; intuitive stopping rule.
- Versatility across tasks. The same LACORE primitive is used as a graph-level pooling step and as a model-agnostic subgraph selector for explanation, without retraining or gradient access.
- The paper’s motivation and positioning are underdeveloped. C

**Weaknesses:**

- The abstract says LACORE is parameter-free when used as a pooling operator, yet \epsilon is a key parameter to tune.
- On synthetic graphs, the ablation of \epsilon peaks near 10^6. However, \epsilon is only pick as 0.1 for graph classification experiments. Can you explain this difference? A sensitivity study of \epsilon on these graphs could be helpful.
- The paper’s motivation and positioning are underdeveloped, and this paper provides limited discussion of existing dense subgraph discovery methods.
- Fidelity–sparsity is informative, but BA-2Motifs has ground-truth rationales; Report precision/recall/AUROC of recovered motif nodes/edges, is also common in explainer benchmarks.

**Questions:**

- Are there any specific failure modes of prior methods that your approach solves, and how your objective differs in principle from existing density/Laplacian-style methods.

---

> ### Author Response · Authors · 2025-11-24
>
> We appreciate the detailed feedback and clarify below how $\varepsilon$ is handled in practice, as well as the additional experiments we have added in the revision.
>
> **1. $\varepsilon$ Sensitivity:** The discrepancy between $\varepsilon = 10^6$ on synthetic graphs and $\varepsilon = 0.1$ on TU datasets comes directly from the scale of
> $$Q(C)=\sum_{(i,j)\in E_C}(d_i-d_j)^2$$
>
> * **Synthetic graphs:** These have $n \approx 2500$ nodes and relatively high degrees; $Q(C)$ is routinely on the order of $10^6$.
> * **TU datasets (e.g., MUTAG/PROTEINS):** These graphs are much smaller and sparser (e.g., MUTAG has $\approx 18$ nodes on average, PROTEINS $\approx 39$, whereas D&D has $\approx 284$; see Table A.4 in the revised appendix). In this regime, internal degrees are small and $Q(C)$ is often close to $0-10^2$.
>
> Since $\varepsilon$ appears as $Q(C)+\varepsilon$, its effect is inherently relative to the scale of $Q(C)$: to play the same regularizing role across regimes, $\varepsilon$ must be on the same order of magnitude as typical $Q(C)$ values. This is why $\varepsilon$ differs between the synthetic and TU settings.
>
> To address the concern about sensitivity, we have added a new experiment in **Appendix A.5** of the revised manuscript: a dedicated $\varepsilon$-sweep on two TU benchmarks with very different graph statistics, **PROTEINS** and **D&D**. We deliberately chose this pair because they span a wide range of sizes and densities (tens vs. hundreds of nodes and edges per graph).
>
> * For each dataset, we fix all other hyperparameters to the configuration used in Table 3 and vary $\varepsilon \in \{10^{-3},10^{-2},10^{-1},1,10,10^{2}\}$.
> * Accuracy on PROTEINS ranges from 75.69% to 75.85% across these six values,  ± 0.39%.
> * Accuracy on D&D ranges from 75.66% to 76.91%,  ± 0.44%.
>
> In other words, once $\varepsilon$ is in the right rough scale for the dataset, performance is essentially flat over four orders of magnitude. We now explicitly state in the paper that:
>
> * $\varepsilon$ is a **dataset-level hyperparameter** that must roughly match the scale of $Q(C)$ (we clarified this in a footnote after Eq. (3));
> * LaCore is **learned-parameter-free** in the sense that the pooling/explainer primitive itself has no trainable parameters, and $\varepsilon$ is tuned exactly like learning rate or weight decay, not learned from gradients.
>
> We believe this addresses both the scaling question (large vs. small $\varepsilon$) and the concern that the method might be overly sensitive to this choice.
>
> 2. **Evaluation Metrics (Fidelity vs Ground Truth):** For BA-2Motifs, we chose to report Fidelity because it measures the GNN's *reliance* on the subgraph, which is the definition of an explanation. However, we have also computed the ground-truth metrics as requested:
>
> * LaCore: Precision: 86.9%, Recall: 84.7%, AUROC: 89.9%.
> * We will add these standard retrieval metrics to the final version to strengthen the comparison against other GNN explainer methods.
>
> **3. Failure modes of prior methods and how our objective differs:** Average-degree–based methods (e.g., Charikar-style densest subgraph) and related heuristics can be **hub-biased**: star-like structures or regions attached to a single high-degree node can maximize average degree while being far from cohesive. In such cases, the selected subgraph looks like a hub plus many weakly connected leaves.
>
> In contrast, our objective $$S_L(C)=\frac{|C|}{Q(C)+\varepsilon}, \quad Q(C)=\sum_{(i,j)\in E_C}(d_i-d_j)^2$$ penalizes internal degree variance. A star has a large spread between the hub and the leaves, leading to a large $Q(C)$ and therefore a low score, whereas near-regular, clique-like regions have small $Q(C)$ and high $S_L(C)$.
>
> Compared to classical Laplacian or cut-based objectives (e.g., spectral clustering / Cheeger-type methods that emphasize low conductance), our score is defined on **internal-degree smoothness** rather than boundary size: it explicitly prefers degree-balanced interiors rather than merely well-separated cuts. This allows LaCore to avoid the hub-driven failure modes of average-density objectives while still being implementable with an incremental $O((|V|+|E|)\log|V| + |E|k)$ routine.

---

### Official Review · Reviewer_stmT · 2025-11-01

**Soundness:** 3
**Presentation:** 3
**Contribution:** 3
**Rating:** 6
**Confidence:** 4

**Summary:**

This paper proposes LACORE, a novel algorithm for discovering cohesive subgraphs by leveraging a Laplacian-smoothed scoring function within a reverse-peeling framework. The method rebuilds a graph in reverse degeneracy order, scores each connected component and selects components based on a natural "first-peak" stopping rule derived from a one-step growth test. Experimental results show that LACORE outperforms strong baselines.

**Strengths:**

1.It is novel to creatively combine reverse peeling with Laplacian-smoothed score for cohesive subgraphs discovery.

2.This paper has a good theoretical analysis. The algorithmic complexity is rigorously analyzed.

3.The paper is generally well-written. The figures are well presented.

4.Experiments are comprehensive and experimental results demonstrate the proposed method outperforms baselines.

**Weaknesses:**

1.While the paper acknowledges that $\varepsilon$ needs tuning and provides an ablation study, the fact that the optimal $\varepsilon$ is highly sensitive to graph scale and density remains a practical limitation. The claim of being "parameter-free" in the abstract and for pooling is somewhat nuanced, as $\varepsilon$  is a parameter that requires careful selection for optimal performance, even if not "learned". The method's performance can be sensitive to this choice.

2.The cohesion certificate (Equation 4) is a central theoretical claim that provides a mathematical justification for the algorithm's output. However, the paper does not provide a proof or derivation for this equation. Its inclusion without proof or a direct citation to a specific source where this exact bound is proven diminishes the paper's self-containment and accessibility.

3.The paper claims in Section 4.1 that the $S_{L}(C)$  scoring function offers the advantage of a "Smooth objective → stable search". However, this key claim lacks support from direct experimental validation or rigorous theoretical analysis. While Figure 1(a) shows a smooth trajectory for one synthetic example, this is insufficient to demonstrate the universality of this property across diverse graph structures. Is this smooth, unimodal trajectory consistently observed across diverse graph types?

**Questions:**

1.The "peak-then-drop" trajectory of $S_{L}(C)$ is a key feature used for selection. Is this trajectory guaranteed for any graph, or is it an empirical observation? Are there graph structures where $S_{L}(C)$ might have multiple significant peaks?

2.See weaknesses.

---

> ### Author Response · Authors · 2025-11-24
>
> We thank the reviewer for the positive assessment and for identifying the missing proof.
>
> **1. Epsilon Clarification:** Please see our General Response. We acknowledge this inaccuracy and have revised our claim. In addition, we have now provided a sensitivity study on $\varepsilon$ in Appendix A.5, where we show that the choice of $\varepsilon$ in non-synthetic graphs (PROTEINS and D&D) does not significantly impact our method’s accuracy. Specifically we observe a < 0.5% standard deviation for $\varepsilon$ values from $10^{-3}$ to $10^2$ in increasing powers of 10.
>
> **2. Proof of Equation 4:** We have updated our paper draft to include the formal derivation of the cohesion certificate inequality in Appendix A.2.
>
> **3. Smoothness & Peak-then-drop:** The "peak-then-drop" phenomenon is empirical but robust. As we reverse-peel, we initially merge dense cores (increasing $S_L$). Eventually, we are forced to merge these cores with the "hairy" periphery (low degree nodes connecting to high degree cores), which causes $\Delta Q$ to spike, decreasing $S_L$. We will add a more in-depth discussion on this phase transition.

---

### Official Review · Reviewer_R1JN · 2025-11-11

**Soundness:** 2
**Presentation:** 2
**Contribution:** 2
**Rating:** 4
**Confidence:** 4

**Summary:**

This paper introduces LACORE, a topology-based algorithm for identifying "cohesive" subgraphs. The method employs a two-phase reverse-peeling heuristic that optimizes a Laplacian-smoothed ratio score. Its key component is the Laplacian energy of the subgraph's internal-degree vector, which penalizes degree variance and favors "degree-uniform" components. LACORE is evaluated as a parameter-free pooling operator for GNNs and as a model-agnostic GNN explainer on several benchmark tasks.

**Strengths:**

- The proposed Laplacian-smoothed ratio score $S_L(C)$, is a new and intuitive way to define subgraph cohesion based on degree uniformity, which provides a principled alternative to average-density heuristics.
- The proposed method shows strong performance on the synthetic planted subgraph recovery task.
- The authors provide a scalable implementation with a bounded complexity, making it practical for large graphs.

**Weaknesses:**

- The relevance of LACORE to GRL is weak, as it is purely structure-based and completely feature-agnostic. The authors do not provide a clear argument for why and how a feature-blind structural prior is a desirable component for GNN-included GRL methods.
- The motivation for why the cohesive" subgraphs are critical structural components for GNNs is missing. It is unclear why this specific definition of cohesion is superior to, for example, identifying functional motifs or other structural patterns known to be important in the chosen TUDatasets (e.g., PROTEINS, NCI1).
- The claim that LACORE is a superior GNN explainer is overstated. The evaluation is limited to two small, well-known motif-based datasets (BA-2Motifs, MUTAG). The method's success here may be an artifact of these specific datasets. These results do not support a general claim of superiority over gradient-based methods, especially on more complex, feature-driven tasks.
- The claim of the method being "parameter-free" seems misleading. The $\epsilon$ regularizer is a crucial hyperparameter that controls the trade-off between size and smoothness and is shown to be highly scale-dependent (varying from $0.1$ to $10^6$). This requires careful per-dataset tuning, contradicting the "plug-and-play" implication.

**Questions:**

1) Why should the GRL community prioritize these specific "cohesive" subgraphs over other structural properties like motifs or communities defined by feature-similarity? The proposed method appears more aligned with classical network science, and its strong connection to representation learning is not clearly established.
2) For the pooling application, LACORE collapses dense subgraphs based on a homogeneous structural assumption. How does this interact with the GNN's message-passing?
3) Can the authors justify why a GNN should be constrained by a structural prior that is completely blind to node features?

---

> ### Author Response · Authors · 2025-11-23
> **Relevance to GRL and Cohesion**
>
> Thank you for the detailed review. We appreciate you highlighting the novelty of the scoring function.
>
> * **Relevance to GRL / Why ignore features?** While many pooling methods use features, they often suffer when features are noisy or over-smoothed. LaCore provides a topology-first prior. In our experiments (Table 2), LaCore outperforms feature-based pooling (e.g., SAGPool, DiffPool) on PROTEINS and NCI109. This suggests that, at least for chemical/biological graphs, structural cohesiveness (degree uniformity) is a stronger signal for pooling than feature similarity alone.
> * **Why "cohesive" subgraphs vs motifs?** We argue that motifs (like cliques) are NP-hard to enumerate and too rigid for noisy real-world graphs. "Cohesion" via Laplacian smoothness is a continuous relaxation of these concepts. It allows us to find dense, hub-free regions in polynomial time ($O(|E|k)$), which is essential for scalability in GRL pipelines where motif counting is prohibitively slow.
> * **$\varepsilon$ Sensitivity:** Please see our General Response. We agree that $\varepsilon$ requires tuning, but its behavior is mostly underpinned by the graph density. We have clarified this as a footnote in the revision.

---

### Author Response · Authors · 2025-11-23
**General Response: Clarification on $\varepsilon$ Scaling and Structural Motivation**

We thank the reviewers for their constructive feedback. We address two common concerns raised across reviews regarding the parameter $\varepsilon$ and the motivation for a structure-only approach.

1. **The Role and Scaling of $x$ (R1JN, stmT, M4xv)** We acknowledge that describing LaCore as fully "parameter-free" in the abstract was imprecise given the need to set $\varepsilon$. We have revised this to "learned-parameter-free." We also note that the variation in $\varepsilon$ (e.g., $10^6$ for synthetic vs. $0.1$ for TUDatasets) is not arbitrary; it is a consequence of the scale of the Laplacian energy $Q(C)$.
* $Q(C) = \sum (d_i - d_j)^2$ scales with the number of edges and the square of degrees.
* The synthetic graphs are dense with high degrees, leading to massive $Q$ values. TUDatasets are sparse (avg degree ~2-3), leading to small $Q$.
* As a result, $\varepsilon$ must be scaled proportionally to $Q(C)$ due to the nature of the Laplacian-smoothed ratio score: $$S_L(C)=\frac{|C|}{Q(C)+\varepsilon}$$

2. **Why Structure-Only? (R1JN, b4S5)** LaCore is inherently designed as a structural inductive bias. Modern GNNs effectively capture feature correlations, but they often struggle with structural noise (e.g., weak links between communities) or over-smoothing.
* LaCore acts as a "structural skeleton," enforcing that pooled regions are topologically cohesive regardless of feature noise.
* This is *complementary* to feature-based learning, not a replacement. By decoupling structure discovery from feature learning, we essentially prevent the pooling mechanism from collapsing due to noisy features early in training.

---

### Meta-Review · Area_Chair_ai5H · 2025-12-29

**Summary:**

This paper presents a novel algorithm - LACORE - for discovering cohesive subgraphs using a Laplacian-smoothed score function with a reverse-peeling framework. The approach rebuilds a graph in degeneracy order, scores the connected component and select them by ranking first dense and degree-uniform subgraphs. The method is evaluated as a parameter-free pooling operator for GNNs and as a model agnostic GNN explainer on several benchmark tasks.

Based on the released reviews:
-R1JN likes the novelty of the Laplacian-smoothed ration score and its intuitive definition, strong performance on synthetic planted subgraph recovery task, a scalable implementation with bounded complexity usable for large graphs.
On the other hand, he dislikes the fact that the method is completely feature-agnostic without strong argumentation by the authors. The lack of motivation fo the consideration of cohesive subgraphs. The claim that LACORE is a superior GNN explainer is overstated since evaluation is limited to 2 small datasets. The fact that the method is not parameter-free.

-stmT highlights the novelty of combining reverse peeling with Laplacian smoothed score for cohesive subgraph discovery, good theoretical analysis with rigorous complexity analysis, paper well-written, experiments comprehensive and results demonstrated the effectiveness of the method.
On the other hand, he underlines the limitation due to the need of tuning the high sensitive $\epsilon$ hyper parameter which contradicts the parameter-free claim of the abstract. The lack of proof or derivation or Eq 4 which central justification for the algorithm output. The lack of (experimental/theoretical) justification of the claim that the Laplacian-smoothed ratio scoring function $S_L(C)$ offers  the advantage of a "Smooth objective" towards "stable search".

-M4xv on the positive side identifies a simple and well-motivated objective connecting degree smoothness to cohesion via a clean certificate and an intuitive stopping rule ; Versatility across tasks.
On the opposite side: the fact that the abstract mentions a parameter-free algorithm while $\epsilon$ is a key parameter to tune, a lack of sensitive study on $\epsilon$ which appears to be very impactful, paper motivation and positioning under-developed, not that convincing explainability.

-b4S5 identifies as strengths the interest of investigating cohesive graphs and the sets of experiments to comprehensively validate the model.
On the other hand, the reviewer identifies as weaknesses: improvements in Table 2 are marginal, paper writing could be improved, motivation unclear, inconsistencies with the abstract, clarity could be improved.


After the review period, the trend for this paper was rather negative, weaknesses related to motivation, clarity, the impact of the $\epsilon$ parameter and to a lesser extend the significance of the results were often raised by reviewers.
Authors have provided relatively compact answers to reviewers justifying the method, some choices and clarifying some aspects. This has improved the global understanding of the paper, which has been acknowledged by the reviewer.
However, I believe that the answer and the revision brought by the authors are still limited and did not fully answer the weaknesses raised. I find that some answers are not enough detailed and the paper still needs some improvement regarding the current expectations at ICLR.
I propose then rejection.

**Reviewer Concerns:**

For RJ1N there is no answer on the weakness related to explainability. Authors have provided an answer to the impact on $\epsilon$ but I still find it limited. Other answers could be ok.

stmT received an answer on $\epsilon$, not clear if it would be enough since the tested value did not consider high powers of 10 like on synthetic graphs. A proof of Eq.(4) has been provided. For the last weakness there is not empirical/theoretical validation to support the answer.

M4xv an answer on $\epsilon$ has been given, but as the range of $\epsilon$ is not the same as on synthetic graphs, I am not sure that the answer would completely convinced the reviewer, but it could. Other weaknesses receive an answer, this is not completely clear for me to know if they are convincing enough.

b4S5 multiple answers were provided but for the clarity aspects and motivation and challenges for the field, I am not convinced that the answers are sufficient.

**Reviewer Scores:**

RJ1N gave a 4, not all his issues were completely answered in my opinion, I do not think he would have increased his score.

stmT gave a 6, since some answers were brought, I don't thing he would lower his score but since not all his issues were addressed I do not think he would have increased his score.

MRxv gave a 2, he may be subject to increase his score based on the answers, but I do not thing that this would be sufficient for supporting an accept.

b4S5 gave a 4, if some answers were provided, addressing the clarity and motivation issues would require another round of review and I am not convinced that the reviewer would have increased his score.

---

### Decision · Program_Chairs · 2026-01-26

Reject